| Open Peer Review | Computational Biology | Methods and Protocols

# A deep learning method for predicting the minimum inhibitory concentration of antimicrobial peptides against *Escherichia coli* using Multi-Branch-CNN and Attention

Jielu Yan,[1] Bob Zhang,[1] Mingliang Zhou,[2] François-Xavier Campbell-Valois,[3,4,5] Shirley W. I. Siu[6]

**ABSTRACT**  Antimicrobial peptides (AMPs) are a promising alternative to antibiotics to combat drug resistance in pathogenic bacteria. However, the development of AMPs with high potency and specificity remains a challenge, and new tools to evaluate antimicrobial activity are needed to accelerate the discovery process. Therefore, we proposed MBC-Attention, a combination of a multi-branch convolution neural network architecture and attention mechanisms to predict the experimental minimum inhibitory concentration of peptides against *Escherichia coli*. The optimal MBC-Attention model achieved an average Pearson correlation coefficient (PCC) of 0.775 and a root mean squared error (RMSE) of 0.533 (log μM) in three independent tests of randomly drawn sequences from the data set. This results in a 5–12% improvement in PCC and a 6–13% improvement in RMSE compared to 17 traditional machine learning models and 2 optimally tuned models using random forest and support vector machine. Ablation studies confirmed that the two proposed attention mechanisms, global attention and local attention, contributed largely to performance improvement.

**IMPORTANCE**  Antimicrobial peptides (AMPs) are potential candidates for replacing conventional antibiotics to combat drug resistance in pathogenic bacteria. Therefore, it is necessary to evaluate the antimicrobial activity of AMPs quantitatively. However, wet-lab experiments are labor-intensive and time-consuming. To accelerate the evaluation process, we develop a deep learning method called MBC-Attention to regress the experimental minimum inhibitory concentration of AMPs against *Escherichia coli*. The proposed model outperforms traditional machine learning methods. Data, scripts to reproduce experiments, and the final production models are available on GitHub.

**KEYWORDS**  minimum inhibitory concentrations, deep learning, regression, antimicrobial peptides, drug discovery

Antimicrobial peptides (AMPs) are a class of molecules found in the innate immune system of living organisms (1, 2). Many of them have shown broad-spectrum activity, rapid killing action, and remarkable potency against bacteria, yeasts, fungi, viruses, and parasites. Some AMPs also exhibited promising *in vivo* activity against multidrug-resistant pathogens (3). Because their mechanism of action differs from that of general antibiotics, they are thought to lead to bacterial resistance only to a limited extent and thus represent an option for the next generation of antibiotics (4).

Biological experiments have been the mainstream method to discover new AMPs from isolates of living organisms. However, these wet-lab procedures are both labor-intensive and cost-intensive. With recent advances in the field of artificial intelligence and the availability of public AMP databases, the scientific community has made intense efforts to develop more accurate prediction tools to aid in the detection and design of

Address correspondence to Bob Zhang, bobzhang@um.edu.mo, or Shirley W. I. Siu, shirley.siu@usj.edu.mo.

The authors declare no conflict of interest.

See the funding table on p. 16.

AMP sequences. Based on our recent review of the computational studies on machine learning and deep learning of AMPs (5), we found that most of the prediction methods only focused on the binary classification of AMPs.

Despite this, there are only a few studies on the quantitative prediction of AMPs. Nagarajan et al. (6) trained a Bi-LSTM regression model on 501 sequences against *Escherichia coli* from the YADAMP database that predicts a scaled minimum inhibitory concentration (MIC) value in the range of 0–1 for a peptide. Although strong AMPs were found with the model, the correlation between the experimental measurement and the prediction was very poor. Dean et al. (7) developed a peptide generation framework, PepVAE, based on a variational autoencoder to sample new sequences from different regions of the learned latent space and on a gradient boosting regressor to predict the MIC of new sequences. The regressor model was based on AMPs for *E. coli*, obtained by comparing eight different models, including convolution neural network (CNN), elastic net (8), gradient boosting (9), kernel ridge (10), Lasso (11), random forest (RF) (12), light gradient boosting machine (13), and extreme gradient boosting (14). The best model was achieved by the gradient boosting algorithm with an $R^2$ value of 0.73 and a root mean squared error (RMSE) of 0.5.

Instead of predicting MIC values, some works exploited the MIC values to create positive and negative data sets for AMP classification. For example, in the studies by Vishnepolsky et al. (15) and Soylemez et al. (16), the authors partitioned AMPs with experimentally known MIC values against a bacterial target into the positive sets (with low MIC values of <25 μg/mL) and the negative sets (with high MIC values of >100 μg/mL). Subsequently, Vishnepolsky et al. constructed a semi-supervised machine learning method with density-based clustering algorithm (DBSCAN) using physical-chemical features as input. Their *E. coli* classification model achieved an accuracy of 80% on the validation set. On the other hand, Soylemez et al. (16) utilized RF and achieved an accuracy of 98% and 95% for the Gram-negative and Gram-positive data sets, respectively.

In addition to predicting AMP activity, it is crucial to assess their potential therapeutic properties, such as anticancer activity, in order to fully understand their potential as novel drug candidates. In this regard, Chen et al. (17) developed xDeep-AcPEP, a powerful regression model that predicts the anticancer activity of peptides for six tumor cells, including breast, cervix, colon, lung, skin, and prostate cancers. They used a multitask learning strategy to train the model with combined data for different tumors, to complement the lack of data for tumor-specific peptides. The best models with the applicability domain defined obtained an average Pearson correlation coefficient (PCC) of 0.8086.

Accurate prediction of the MIC of AMPs remains challenging due to limited data, variations in activity measurements across different experimental methods and conditions, and the presence of noise in the data. As a step toward accurate quantitative prediction of AMP activity, we present MBC-Attention, a regressive model designed to predict the MIC value of AMPs against *E. coli* with greater precision. Here, a predicted MIC value refers to the antimicrobial potential of a peptide that is not specific to particular experimental conditions. One should be aware that different MIC values can be obtained for the same AMP under different experimental conditions such as growth media, temperature, pH, ionic strength, and salt type. Although it is noteworthy that standard media is often used to measure MIC in a given bacterial specie, there are other experimental parameters such as the purity of the peptide that can explain diverging reported MIC values for a given AMP. The reason for learning the overall potential instead of a condition-dependent MIC value in the experiment is mainly because there is no comprehensive data set for AMPs measured under many different conditions. When multiple entries for an AMP exist, the average MIC is calculated as the prediction target. This approach provides a reasonable estimate of the antimicrobial potential of the sequence. The same approach was adopted in a previous study that developed the PepVAE regression model (7). In this work, MBC-Attention is built

on the Multi-Branch-CNN (MBC) architecture (18), a deep learning approach featuring multiple parallel CNN branches at the input level. By concatenating the predictions from multiple CNNs, MBC-Attention generates a binary prediction at the output level. Originally developed for classifying ion channel binding peptides, we adapted MBC for regression by incorporating attention mechanisms, resulting in improved performance. Two attention mechanisms were introduced into MBC: attention-local, which enables focusing within each CNN branch, and attention-global, which facilitates connections across the different CNN branches.

The entire data set was randomly divided into training, validation, and test sets. The test set served as the hold-out set for the final model evaluation, while the training and validation sets were used for model building and optimization. To ensure a reliable estimate of model performance, we generated three sets of training, validation, and test sets by repeating the data splitting procedure. For each prediction method, training was performed three times and the method was evaluated based on the average PCC of the three validation results to determine the best model with optimal parameters. PCC is a measure of linear correlation between two data sets that reflects their degree of correlation. The metric with a value of −1 indicates a completely negative correlation, 0 indicates no correlation, and 1 indicates a completely positive correlation. In the comparative study, the proposed MBC-Attention model was compared with TML17 [the best model selected from 17 traditional machine learning (TML) methods using an automated routine] and two other widely used methods RF (12) and SVM (19) with optimally tuned hyperparameters. Furthermore, to evaluate whether the proposed MBC-Attention method performed better than MBC on classification tasks, we compared the two methods using the ion channel peptide prediction data sets from our previous work (18). To confirm the importance of the designed components in MBC-Attention, we conducted a series of ablation tests, including MBC alone, MBC with attention-global, and MBC with attention-local.

## RESULTS

### Single feature selection

Prior to training a machine learning model, the peptide sequences in the data set were converted into numerical values called *features*, which represent different properties of the sequence at the peptide level or at the amino-acid level. To select the most informative feature types for encoding AMP sequences, a total of 4,481 different feature types were examined. These include the "OneHot" feature type, 53 feature types that were generated by different feature encoding methods [e.g., "AAC," "CKSAAP" (20), and "KSCTRiad" (21)], and 4,427 feature types generated by the Pseudo K-tuple Reduced Amino Acids Composition (PseKRAAC) encoding method (22). For each feature type, we constructed an RF using 400 trees and computed its predictive performance in terms of PCC. Each experiment was repeated three times and the average of PCC scores in the three validation sets was computed for each feature type. The purpose of this step is to identify the most informative feature types for the MBC-Attention model construction.

As shown in Fig. 1, about half of the 4,481 feature types demonstrated average PCC scores between 0.55 and 0.743, and 205 of these feature types (or the top 4.5% of the feature types) had model performance within 1 standard deviation of the best model. The most informative feature types are mainly from the PseKRAAC method with different clustering types and encoding parameters. Several QSOrder and some CKSAAP feature types were also found informative. The error bar plot of the top feature types' PCC scores shows that there are no statistically significant differences in performance among the different top feature types, as evidenced by the overlapping 1 standard deviation range.

### Best-*K* feature combination selection

In order to optimize the predictive accuracy of MIC values of AMPs against *E. coli*, it is crucial to identify the ideal combination of feature types to be learned. To this end, we

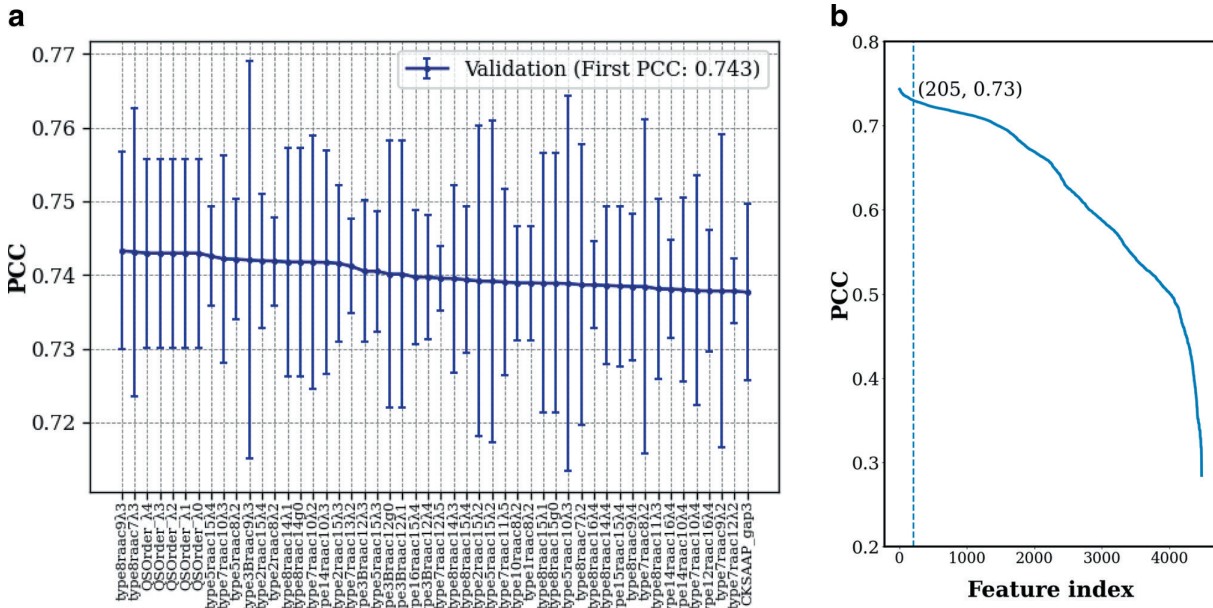

**FIG 1** Prediction performance of single-feature RF models with respect to PCC: (a) The 50 most informative features and (b) all 4,481 features. The top 205 features having PCC scores within 1 standard deviation of the best model is to the left of the dashed line.

evaluated the feature combinations best-1, best-2, and so on, using MBC-Attention. The best-$K$ feature combination here refers to the first $K$ feature types that were ranked based on their individual performance in the aforementioned single feature selection. The MBC-Attention model for the best-$K$ feature combination was trained using the hyper-parameters (CNN layer = 2, filter = 64, dropout = 0.4, learning rate = 0.0001 and patience = 20, decay rate = 0.9, and decay steps = 100) that were previously identified for the MBC model in our previous work (18). Training an MBC-Attention model was feasible up to the best-26 feature combination. For more feature types requiring larger architectures with more CNN branches, we were unable to train due to computational constraints.

Figure 2 shows the performance of the MBC-Attention models with varying number of feature combinations, indicated as best-1 to best-26. These models were ranked in descending order by their average PCC scores on the validation sets. The PCC scores of the models in the test sets are also shown, confirming that our model selection based on the validation performance is appropriate. According to the results, the best-1 to best-15 models consistently demonstrated stable performance in the training, validation, and test experiments. Conversely, the best-16 to best-26 models show poor performance with substantially reduced PCC scores and exhibit large standard deviations. Although the best-16 and best-17 models produced reasonable results in validation, they show exceptionally poor predictions in the test sets for unknown reasons. With the exception of the best-16 and best-17 models, all models in the test experiments show a similar performance trend as in the validation experiments, indicating that model selection based on the validation results is reliable.

As a result, the best-14 feature combination, which yielded an average PCC of 0.748, was selected for final model optimization. The feature types included in best-14 are listed in Table 1. It is worth noting that most of the best feature types are based on the PseKRAAC and QSOrder encoding methods. Feature types from the same method mainly differ in their parameters such as the number of reduced amino acid clusters and the λ value, the latter being the positional distance between two amino acids along the sequence for correlation analysis. The small λ values between 0 and 4 from the best-14 feature combination indicate that closely adjacent amino acids are likely to be more relevant for the antimicrobial function of peptides.

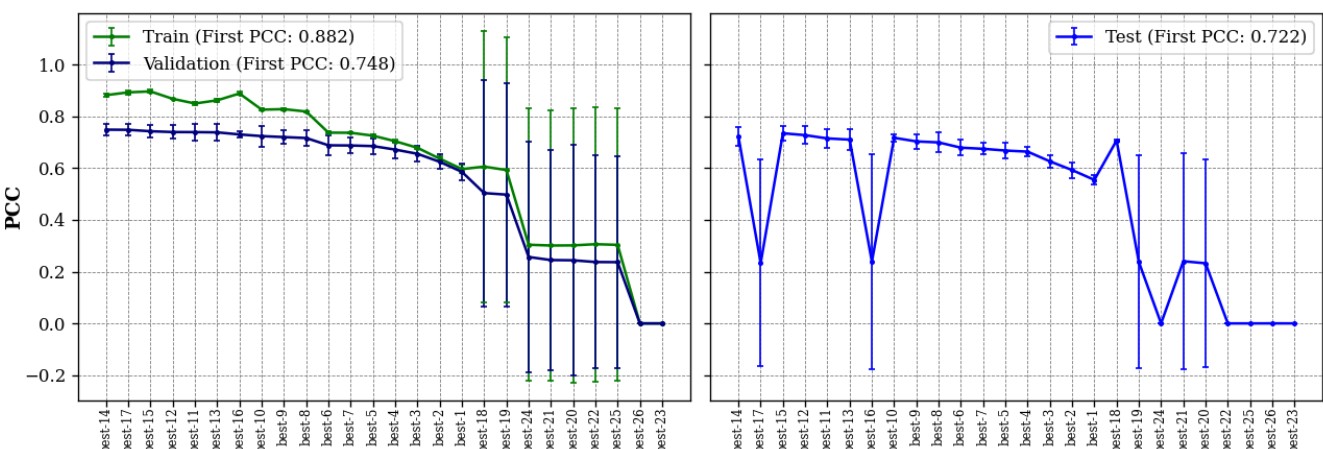

**FIG 2** Training, validation, and test performances of the best-1 to best-26 models using MBC-Attention in feature combination selection. The experiments were conducted three times, and the average PCC scores are presented with error bars indicating 1 standard deviation from the mean.

**TABLE 1** The best-14 feature types determined to be the most effective to use with MBC-Attention for predicting the MIC of a peptide against *E. coli*[a]

| Feature groups | Type | Type description | RAAC no. of clusters | $\lambda$ | No. of features |
|---|---|---|---|---|---|
| PseKRAAC | 8 | Grantham distance matrix (saturation) (23) | 9 | 3 | 81 |
| PseKRAAC | 8 | Grantham distance matrix (saturation) | 7 | 3 | 49 |
| QSOrder | – | Quasi-sequence-order (24) | – | 4 | 343 |
| QSOrder | – | Quasi-sequence-order | – | 3 | 343 |
| QSOrder | – | Quasi-sequence-order | – | 2 | 343 |
| QSOrder | – | Quasi-sequence-order | – | 1 | 343 |
| QSOrder | – | Quasi-sequence-order | – | 0 | 343 |
| PseKRAAC | 5 | BLOSUM50 matrix (25) | 15 | 4 | 225 |
| PseKRAAC | 7 | Metric multi-dimensional scaling (MMDS) (26) | 10 | 3 | 100 |
| PseKRAAC | 5 | BLOSUM50 matrix | 8 | 2 | 64 |
| PseKRAAC | 3B | Whelan and Goldman (WAG) matrix (27) | 9 | 3 | 81 |
| PseKRAAC | 2 | BLOSUM 62 matrix (28) | 15 | 4 | 225 |
| PseKRAAC | 2 | BLOSUM 62 matrix | 8 | 2 | 64 |
| PseKRAAC | 8 | Grantham distance matrix (saturation) | 14 | 1 | 196 |

[a]The symbol "-" denotes that no parameters are required for feature generation.

## TML models

To establish the baseline performance for MIC regression of AMPs, we set out to evaluate the performance of TML models. Three models utilizing the best-14 feature combination were constructed: the TML17 model, the optimal RF model, and the optimal SVM model.

TML17 is a routine that automates the evaluation of 17 different TML models (see Table 6) by fivefold cross-validation (CV) on training data sets; the training was done using the default hyperparameters of each model. Finally, the best-performing model was chosen as the TML17 model; its performance was compared to MBC-Attention as will be discussed below.

In addition, we produced the optimally tuned models of RF and SVM by the grid search algorithm. When tuning the RF model, we adjusted the number of trees (ranging from 100 to 2,000 in intervals of 100) and the functions used to determine the maximum number of features when splitting a node ("log2" or "sqrt"). To tune SVM, we experimented with different kernels (including "rbf," "linear," and "poly") and regularization parameters (C = 0.01, 0.1, 1, 10, and 100). We repeated all experiments three times and the results of both training and validation are presented in Fig. 3.

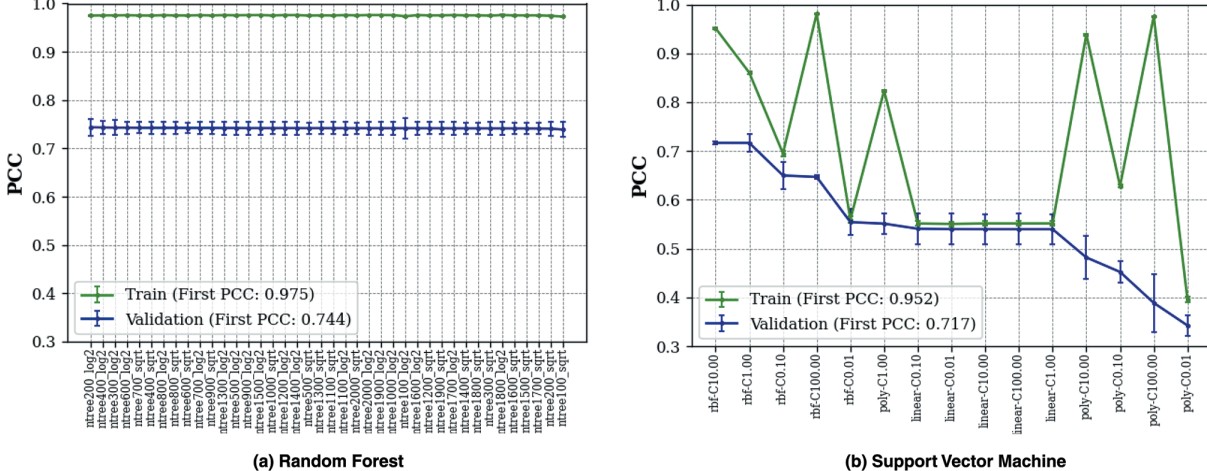

**(a) Random Forest**      **(b) Support Vector Machine**

**FIG 3** Regressive performance of (a) RF and (b) SVM models on the validation sets with different hyperparameters using grid search.

In the case of RF, we observed consistent performance across all models (as shown in Fig. 3a), with no significant differences among them. The model with the highest PCC score in validation was trained using 200 trees and the maximum feature function of "log2." In contrast, SVM models show great sensitivity to the given parameters, leading to substantial variations in their PCC scores in training and validation experiments (Fig. 3b). The best SVM model was trained using an "rbf" kernel and a C value of 10.

As a result, using the best-14 feature combination, the TML17 model, the optimal RF model, and the optimal SVM model achieve PCC scores of 0.737, 0.744, and 0.717, respectively.

To provide a reference, we conducted grid search on RF and SVM using only the best-1 feature type. The resulting average PCCs of the optimal RF and SVM models were 0.738 and 0.715, respectively, on the validation sets. These average scores were slightly inferior to those obtained by their corresponding best-14 models.

## Comparison between MBC-Attention and TML methods

Likewise, the MBC-Attention model was tuned using grid search on various hyperparameters, including number of filters, dropout rate, number of layers, and loss function. Subsequently, we manually fine-tuned the early stopping parameters, such as learning rate, decay rate, decay steps, patience, and monitoring, based on the best hyperparameters obtained from the grid search.

In the comparative study, the optimal MBC-Attention model was compared to three TML models, TML17, the optimal RF, and the optimal SVM models, using the test data sets. Test experiments were conducted three times, similar to the validation experiments, and the mean performance values were reported.

Table 2 displays the full results of the evaluation of the proposed models, which were assessed based on six criteria. Higher values of the average PCC, concordance correlation coefficient (CCC), and $R^2$ scores indicate better model performance (↑), while lower values of the average mean absolute error (MAE), mean squared error (MSE), and RMSE scores suggest better model performance (↓). According to the evaluation results, the MBC-Attention model outperforms all the TML models (TML17, optimal RF, and optimal SVM) on all of the evaluated criteria, with an average PCC of 0.775 in the test data sets. Specifically, MBC-Attention outperforms TML17, RF, and SVM by 5.2%, 5.4%, and 12.2%, respectively.

To investigate the impact of duplicate features in the best-14 feature combination, we conducted a feature elimination experiment. This involved removing all duplicate features before training the MBC-Attention model, using the same procedure as described in Yan et al. (18). However, the reduced-feature version of MBC-Attention

**TABLE 2** Comparative performance of TML17, RF, SVM, MBC-Attention, and reduce-feature MBC-Attention on the test sets[a]

| Algorithm | | PCC ↑ | CCC ↑ | $R^2$ ↑ | MAE ↓ | MSE ↓ | RMSE ↓ |
|---|---|---|---|---|---|---|---|
| | Mean | 0.737 | 0.697 | 0.541 | 0.425 | 0.320 | 0.566 |
| TML17 | Std | 0.020 | 0.022 | 0.028 | 0.012 | 0.017 | 0.015 |
| | Mean | 0.735 | 0.666 | 0.531 | 0.441 | 0.327 | 0.571 |
| RF (200 trees and max features of "log2") | Std | 0.042 | 0.049 | 0.059 | 0.027 | 0.037 | 0.032 |
| | Mean | 0.691 | 0.675 | 0.466 | 0.461 | 0.375 | 0.612 |
| SVM ("rbf" kernel and C of 10) | Std | 0.042 | 0.045 | 0.070 | 0.030 | 0.043 | 0.035 |
| | Mean | **0.775** | **0.738** | **0.588** | **0.402** | **0.284** | **0.533** |
| MBC-Attention | Std | **0.014** | **0.015** | **0.005** | **0.006** | **0.003** | **0.002** |
| | Mean | 0.764 | 0.735 | 0.582 | 0.419 | 0.298 | 0.545 |
| MBC-Attention with reduced features | Std | 0.019 | 0.030 | 0.026 | 0.021 | 0.020 | 0.014 |

[a]The best value for each criterion is indicated in bold.

performs slightly worse than the full-feature model, with an average PCC that is 1.4% lower. Nevertheless, the reduced-feature model still outperforms TML17, RF, and SVM, producing an improved PCC score that is 3.7%, 3.9%, and 10.6% higher than those of the respective models (see Table 2).

It is noteworthy that MBC-Attention shows higher stability than the other three models compared. The standard deviation values obtained for MBC-Attention in all the measures are relatively lower (e.g., for PCC, it is 30–66.7% lower; for RMSE, it is 86.7–94.3% lower) than those for TML17, RF, and SVM. Reducing input features to MBC-Attention also weakened the stability of the model, indicating that the exclusion of certain features may result in the loss of information relevant to the activity prediction of AMPs.

## Data size effect exploration

Deep learning is data-dependent, i.e., it relies on a sufficient amount of data to build a reliable prediction model. When the amount of data is small, deep learning algorithms often perform poorly. Therefore, it is important to decide how much data are needed to train a model to achieve optimal performance and to understand whether the current amount of data is adequate for the training needs. Here, we examined the effects of training data size on model performance for the proposed MBC-Attention model. Four experiments were conducted using training sets of 1,000, 2,000, 3,000, and 3,536 sequences (the full training set), respectively, to train the model. We then evaluated the performance for the model on the same test data set. To ensure statistical significance, each experiment was repeated five times and the results were analyzed.

The PCC scores of all experiments on the test set were visualized in a scattered boxplot, as shown in Fig. 4. The PCC values show a significant improvement in model performance with an increasing training set size. Specifically, the median PCC values exhibit a clear trend of improvement, indicating that the model is able to learn from more training data and make predictions more accurately. The result also shows a clear trend toward convergence, indicating that the current training data size is a reasonable amount to train the MBC-Attention model. Unfortunately, an even larger data set is currently not available to conclude with the optimal data size for this AMP prediction task.

## Ablation study

An ablation study was performed to understand the importance of each component, namely MBC, global attention, and local attention, in the MBC-Attention architecture. As shown in Table 3, MBC alone performs the worst among all four models. The use of local attention increases PCC by 2.97%, and the use of global attention leads to a 3.37% improvement. The combined use of both attention schemes increases MBC performance by 4.59%, highlighting the importance of attention in learning AMP sequences. Additionally, the results show that MBC-Attention has the highest stability among all models, as evidenced by the lower standard deviation in triplicate test experiments.

**TABLE 3** Comparative performance of MBC, MBC with global attention, MBC with local attention, and MBC-Attention (with both global and local) on the test sets[b]

| Algorithm | PCC ↑ (std) | % increase↑ | CCC ↑ | $R^2$↑ | MAE↓ | MSE↓ | RMSE↓ |
|---|---|---|---|---|---|---|---|
| MBC | 0.741 (0.019) | –[a] | 0.692 | 0.528 | 0.451 | 0.336 | 0.580 |
| MBC with global attention | 0.766 (0.017) | 3.37 | 0.743 | 0.587 | 0.411 | 0.293 | 0.542 |
| MBC with local attention | 0.763 (0.016) | 2.97 | 0.735 | 0.581 | 0.419 | 0.298 | 0.546 |
| MBC-Attention | **0.775 (0.014)** | **4.59** | **0.738** | **0.588** | **0.402** | **0.284** | **0.533** |

[a]The symbol "-" indicates that no data is provided for the entry.
[b]The best value for each criterion is indicated in bold.

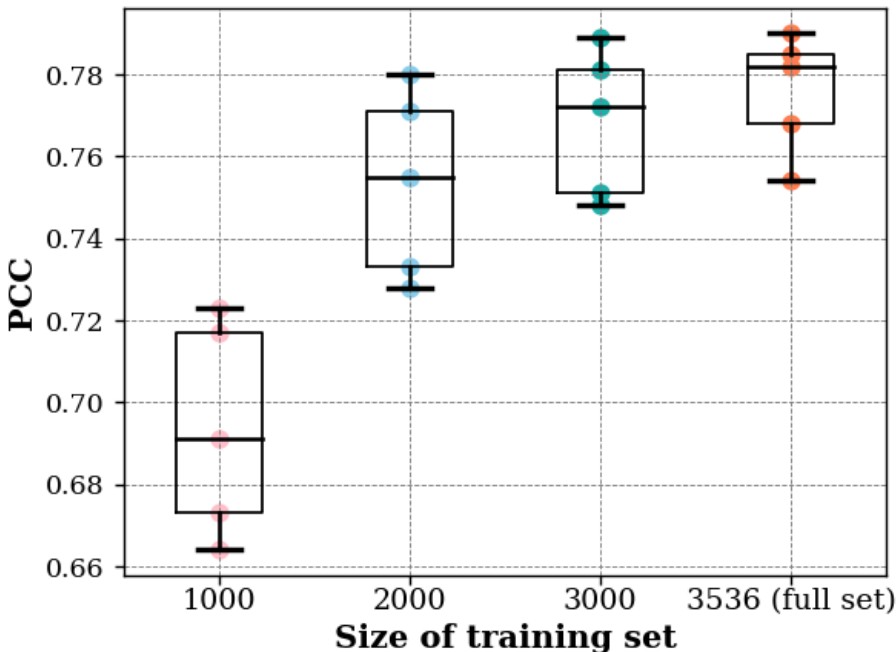

**FIG 4** Effect of varying training set sizes on the performance of MBC-Attention.

## Comparison between MBC-Attention and MBC on ion channel data sets

Given that MBC was originally designed to address the ion channel peptide classification problem, it is interesting to explore whether its derivative, MBC-Attention, can also enhance classification performance.

In reference (18), we proposed three separate models to predict peptides for the sodium (Na-pep), potassium (K-pep), and calcium (Ca-pep) ion channels. For each channel, two different test sets were developed: the test set consists of peptides with moderate similarity to those in the training set, whereas the novel-test set contains peptides with low similarity or dissimilarity to the training peptides. Hence, the novel-test set can be considered a more challenging data set. We integrated the attention-local module in the middle of the 2D convolutional layer and the max-pooling layer for each CNN layer of all CNN branches and the attention-global module at the end of each CNN branch. MBC-Attention has the same hyperparameters as the MBC model used in the work of MBC for Na-pep, K-pep, and Ca-pep.

As shown in Fig. 5, the MBC-Attention outperforms the MBC method for all three channels (Na-pep, K-pep, and Ca-pep) on both the test and novel-test data sets. This finding confirms that attention mechanisms are advantageous also in classification tasks. Further investigation of MBC-Attention in other classification applications is warranted.

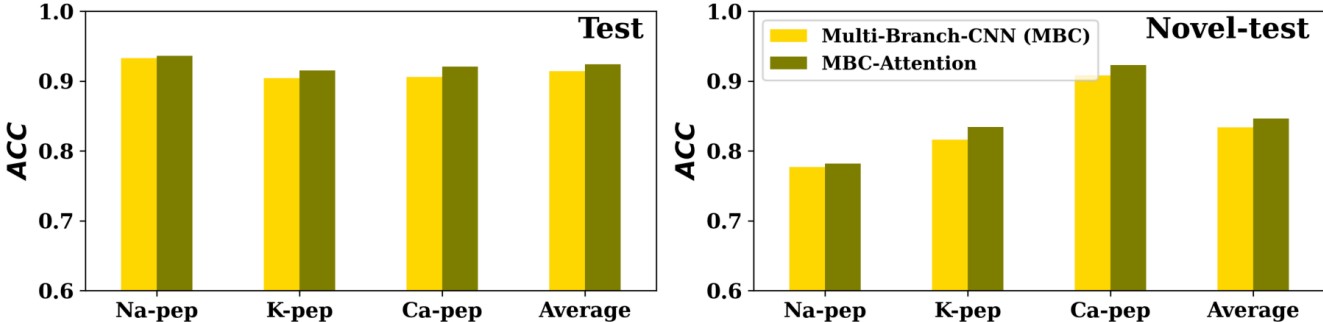

**FIG 5** Comparative accuracy of Multi-Branch-CNN (MBC) and MBC-Attention on test (left) and novel-test (right) sets of Na-pep, K-pep, and Ca-pep predictions.

## DISCUSSION

In this paper, we propose MBC-Attention as a step toward improved prediction of quantitative MIC values of AMPs against *E. coli*. MBC-Attention is built upon the architecture of our previous MBC model (18), which is a multiple CNN branch architecture capable of accurately classifying peptides that interact with ion channels.

The main advantage of MBC is its ability to learn from multiple feature types, even for features having some degrees of redundancy. This is particularly useful in the case of learning amino acid sequences, where there are many sequence encoding methods distinct in terms of the peptide properties that they capture but resulting in predictive models with highly similar performance. This is because the use of TML methods like RF and SVM is ineffective in learning feature types that exhibit certain levels of redundancy leading to a negligible difference in model performance.

In this study, we leverage the MBC architecture for the regression task of quantitative MIC prediction and enhance its predictive capabilities by incorporating two attention mechanisms. The first mechanism, global attention, is employed to facilitate learning by attending to the predictions of CNN branches, each of which takes a different feature type as input. The second mechanism, local attention, is implemented within each branch to enable the model to focus on the most informative features within a feature type.

It is noteworthy that although MBC outperforms the best TML models by only a small margin, incorporating attention mechanisms, greatly enhances MBC's performance. Specifically, the global and local attention mechanisms improve the MBC model's performance by 3.37% and 2.97%, respectively, resulting in a more accurate quantitative model, MBC-Attention, for predicting the AMP activity in *E. coli*. Moreover, we examined the effects of different data sizes, and the result shows a good tendency to converge, indicating that the size of our data set is reasonable for training the proposed MBC-Attention model.

In addition, we showed that the performance of MBC-Attention for classification tasks is improved compared to MBC, evaluated on ion channel peptide classification (18). These results suggest that attention-based approaches may have broader applicability beyond AMP activity prediction.

In the future, we plan to extend the application of the MBC-Attention model to predict the antibacterial activity of peptides against other pathogenic bacterial species. To further improve the accuracy of the prediction of experimental MIC for a sequence, it is crucial to investigate the correlation between antimicrobial activity and different experimental conditions. Future research will also focus on investigating this relationship and improving our predictive capabilities.

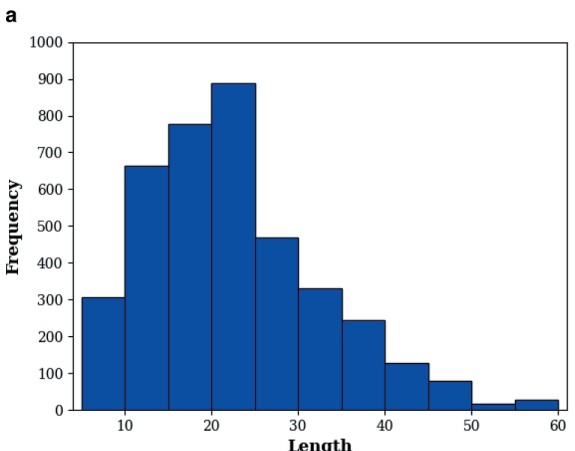
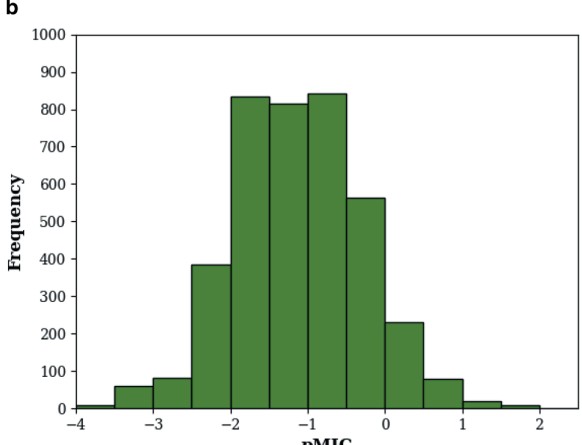

**FIG 6** (Left) The histogram of sequence lengths and (right) pMIC values of the curated AMP data set of 3,929 entries for *E. coli*.

## MATERIALS AND METHODS

### Data set preparation

We downloaded a total of 7,861 data entries of AMP monomers from the DBAASP v3 database (41) with antibacterial activity against *E. coli*. To prepare for machine learning, the data entries with unnatural amino acids ("B," "J," "O," "U," "Z," and "X") in the sequence or entries without experimental MIC values were removed; sequences that are too long (>60 aa) and too short (<5 aa) were discarded. For the remaining entries, their MIC values were converted into the same measurement unit (µM) and then converted to pMIC by Eq. (1). For a sequence that has multiple MIC values, these values were averaged to obtain the mean activity measure for the peptide that represents the overall antimicrobial potential under different experimental conditions. Finally, the processed data set of AMP sequence and MIC for *E. coli* contains 3,929 entries. As shown in Fig. 6, these sequences have a median length of 20 residues and antibacterial activity of 14.8 µM (equivalent to pMIC of −1.17).

To construct and evaluate machine learning models, the data set was randomly partitioned into training, validation, and test sets. The test set comprised 10% of the total data, while the validation set was formed by removing 10% of the remaining data after the test set was separated. To ensure accurate error estimation, the data set-splitting process was repeated three times. Each experiment was conducted with three replicates, and the mean performance values were reported. The information about all downloaded sequences and created sets is shown in Table 4.

### Regressive predictive target

In this work, the negative logarithmic MIC (pMIC) against the bacteria is the regressive predictive target. The pMIC can be represented as follows:

$$pMIC = -\log_{10}MIC \tag{1}$$

### Performance metrics

We used PCC, CCC, MAE, MSE, RMSE, and coefficient of determination ($R^2$) as metrics to measure the performance of the regressive models. Note that the lower the value of MAE, MSE, or RMSE, the better the regression models. Conversely, the higher the value of PCC, CCC, and $R^2$, the better the regressive models.

The formula of PCC or $\rho$ is shown as follows:

**TABLE 4** Summary of the AMP data against the Gram-negative bacteria *E. coli* and data partition for predictive modeling

| Name | Description | No. of entries |
|---|---|---|
| EC of all subtypes | Downloaded from DBAASP (41) (as of August 2021) | 7,861 |
| Whole | After removing incomplete and duplicate entries | 3,929 |
| Test | Randomly select 10% from the whole set | 393 |
| Validation | Randomly select 10% from the whole set after removing test | 354 |
| Train | Data not included in test and validation | 3,536 |

$$PCC = \rho = \frac{\sum_{i=1}^{N}(y_i - \mu_y)(\hat{y}_i - \mu_{\hat{y}})}{\sqrt{\sum_{i=1}^{N}(y_i - \mu_y)^2 \sum_{i=1}^{N}(\hat{y}_i - \mu_{\hat{y}})^2}}, \tag{2}$$

$$\mu_y = \frac{\sum_{i=1}^{N} y_i}{N},$$

$$\mu_{\hat{y}} = \frac{\sum_{i=1}^{N} \hat{y}_i}{N}.$$

where *i* denotes the *i*th sample and *N* is the sample number; *y* represents a true target value and $\hat{y}$ is the predicted target value by the regressive model; $\mu_y$ and $\mu_{\hat{y}}$ represent the average of the true and predicted regressive target values, respectively.

The formula of CCC or $\rho_c$ is shown as follows:

$$CCC = \rho_c = \frac{2\rho\sigma_y\sigma_{\hat{y}}}{\sigma_y^2 + \sigma_{\hat{y}}^2 + (\mu_y - \mu_{\hat{y}})^2}, \tag{3}$$

$$\sigma_y = \frac{\sum_{i=1}^{N}(\mu_y - y_i)^2}{N},$$

$$\sigma_{\hat{y}} = \frac{\sum_{i=1}^{N}(\mu_{\hat{y}} - \hat{y}_i)^2}{N},$$

$$i \in \{1, 2, \ldots, N\}.$$

where $\sigma_y^2$ and $\sigma_{\hat{y}}^2$ denote the corresponding variances of true and predicted regressive target value, respectively.

The formula of MAE is shown as follows:

$$MAE = \frac{1}{N}\sum_{i=1}^{N}|\hat{y}_i - y_i|, i \in \{1, 2, \ldots, N\}, \tag{4}$$

where *i* denotes the *i*th sample of prediction target value of regressive models ($\hat{y}$), *y* represents true target value of regressive models, and *N* is the sample number.

The formula of MSE is presented as follows:

$$MSE = \frac{1}{N}\sum_{i=1}^{N}(\hat{y}_i - y_i)^2, i \in \{1, 2, \ldots, N\}. \tag{5}$$

The formula of RMSE is defined as follows:

$$\text{RMSE} = \sqrt{\frac{1}{N}\sum_{i=1}^{N}(\hat{y}_i - y_i)^2}, i \in \{1, 2, \ldots, N\}. \tag{6}$$

The formula of $R^2$ is shown as follows:

$$R^2 = 1 - \frac{\sum_{i=1}^{N}(\hat{y}_i - y_i)^2}{\sum_{i=1}^{N}(\bar{y} - y_i)^2}, i \in \{1, 2, \ldots, N\}. \tag{7}$$

## Feature encoding of peptide sequences

We examined a total of 4,481 different features to find the optimal combination of AMP sequence encoding methods. The encoding for 4,480 features was generated using the iFeature package (42), while the oneHot feature encoding was implemented ourselves. oneHot produces a $L_{max} \times 20$ feature matrix ($L_{max}$ is the maximum length of all given peptides) which represents the position of each amino acid in the sequence. Among the different features from iFeature, 53 of them were generated by 21 feature types (such as AAC, CKSAAP, KSCTriad, and SOCNumber) using different parameters. But a majority of features (totally 4,427) were from PseKRAAC, which generates encodings based on a set of user-defined parameters including ktuple, g-gap, and λ-correlation. Table 5 shows all feature types and corresponding parameters used in sequence encoding.

## MBC-Attention

MBC is short for Multi-Branch-CNN, which is a deep learning architecture for peptide activity prediction based on amino acid sequences. It was proposed in our previous study of machine learning for ion channel binding peptides prediction (18).

MBC was designed to solve classification tasks. Here, modifications are necessary to address the more challenging regression modeling and to improve predictive performance. The final architecture of MBC-Attention is shown in Fig. 7 which is comprised of multiple CNN branches to extract features from different feature types. Each CNN branch contains a batch normalization layer, a 2D convolutional layer, an attention-local layer, a max-pooling layer, a flatten layer, and a dense layer. After the dense layer in each branch, the 1D outputs are concatenated into a 2D matrix. This matrix is passed to a global attention layer, which identifies how informative is each CNN branch's output. The result of the attention-global layer is then flattened and passed through another dense layer for further processing. Finally, a dense layer with one node is used to generate the final regressive target, which is the negative logarithm of MIC for a given peptide sequence.

The entire network is updated using back-propagation. After the tuning procedure (as detailed below), the optimal MBC-Attention has the following hyperparameters: 14 CNN branches, 1 convolutional layer with 32 filters in each CNN branch, a dense network with 32 nodes, the use of the ReLU activation function for internal nodes, and the sigmoid function for the output node.

To facilitate the extraction of relevant information within each CNN branch and to correlate the outputs from different branches to the predictive target, we propose two attention mechanisms: attention-local and attention-global. Each attention module consists of three elements—a key, a query, and a value. The main idea of attention is to use the similarity of a query and a key to weigh and adjust the value accordingly.

As shown in Fig. 7a, the attention-local module is introduced in each CNN branch. In the attention-local-$i$ layer ($i \in \{1, 2, \ldots, K\}$), the local-$i$ input is a $H \times W \times C$ matrix, where $H$, $W$, and $C$ stand for height, width, and channel, respectively, where $C$ corresponds to the filter number in the previous convolution layer (Fig. 7b). Furthermore, $\otimes$ denotes matrix multiplication and $\oplus$ denotes matrix addition. First, three convolutional layers with blocks of 32 filters whose size is $1 \times 1$ are utilized to generate three equal 3D matrices. They are reshaped into three 2D matrices ($(H \times W) \times C$) to represent the key, query, and value terms. The key is multiplied by the transposed query term to obtain the similarity values, which are processed with the softmax function for each row to obtain

**TABLE 5** The feature types and parameters for encoding peptide sequences[a]

| Feature name | Description | Parameters |
|---|---|---|
| OneHot | One hot encoding | – |
| AAC | Amino acid composition | – |
| DPC | Dipeptide composition (43) | – |
| TPC | Tripeptide composition (44) | – |
| DDE | Dipeptide deviation from expected mean (45) | – |
| GAAC | Grouped amino acid composition | – |
| GDPC | Grouped dipeptide composition | – |
| GTPC | Grouped tripeptide composition | – |
| CTDC | Composition of composition/transition/distribution (46) | – |
| CTDT | Transition of composition/transition/distribution | – |
| CTDD | Distribution of composition/transition/distribution | – |
| CSKAAP | Composition of k-spaced amino acid pairs (47) | $gap \in \{0, 1, 2, 3\}$ |
| CKSAAGP | Composition of k-spaced amino acid group pairs (20) | $gap \in \{0, 1, 2, 3\}$ |
| CTriad | Conjoint triad (21) | $gap \in \{0, 1, 2, 3\}$ |
| KSCTriad | k-Spaced conjoint triad (21) | $gap \in \{0, 1\}$ |
| SOCNumber | Sequence-order-coupling number (48) | $\lambda \in \{0, 1, ..., 4\}$ |
| QSOrder | Quasi-sequence-order (24) | $\lambda \in \{0, 1, ..., 4\}$ |
| PAAC | Pseudo-amino acid composition (49) | $\lambda \in \{0, 1, ..., 4\}$ |
| APAAC | Amphiphilic pseudo-amino acid composition (50) | $\lambda \in \{0, 1, ..., 4\}$ |
| NMBroto | Normalized Moreau-Broto autocorrelation | $nlag \in \{2, 3, 4\}$ |
| Moran | Moran correlation (51) | $nlag \in \{2, 3, 4\}$ |
| Geary | Geary correlation (51) | $nlag \in \{2, 3, 4\}$ |
| | | $ktuple = 2$ |
| | | $subtype \in \{'g - gap', ' \lambda - correlation'\}$ |
| | | $g - gap \in \{0, 1, ..., 9\}$ |
| PseKRAAC | Pseudo K-tuple reduced amino acid composition (22) | $\lambda - correlation \in \{0, 1, ..., 9\}$ |

[a]The symbol "-" denotes that no parameters are required for feature generation.

the local weights. Then, the local weights are multiplied by the value term which are transformed into the same dimension of the input of the attention-local-$i$ module. Finally, matrix addition of the attention-local-$i$ weighted input and the attention-local-$i$ input is performed to adjust the original inputs.

Meanwhile, the attention-global module aims to optimize the information flow from each CNN branch by a self-attention approach as shown in Fig. 7c. To accomplish this, the global-input matrix (obtained from concatenating outputs from all CNN branches) is first transposed and multiplied by itself. The softmax function is then applied to this matrix to generate the branch weights. These branch weights are then multiplied with the global-input matrix, and the resulting matrices are added together to obtain the fitted information of the CNN branches.

## TML17 method

To evaluate the effectiveness of TML algorithms, we followed the TML13 classification method as detailed in our previous work (18) and developed the TML17 method, which is based on 17 different TML methods listed in Table 6. In TML17, it conducts a fivefold CV on the training data set for each algorithm, with the default parameters, and selects the method with the highest PCC score as the final model of TML17 and the method's output as the final prediction of TML17. Implementation of TML17 was carried out using PyCaret (52).

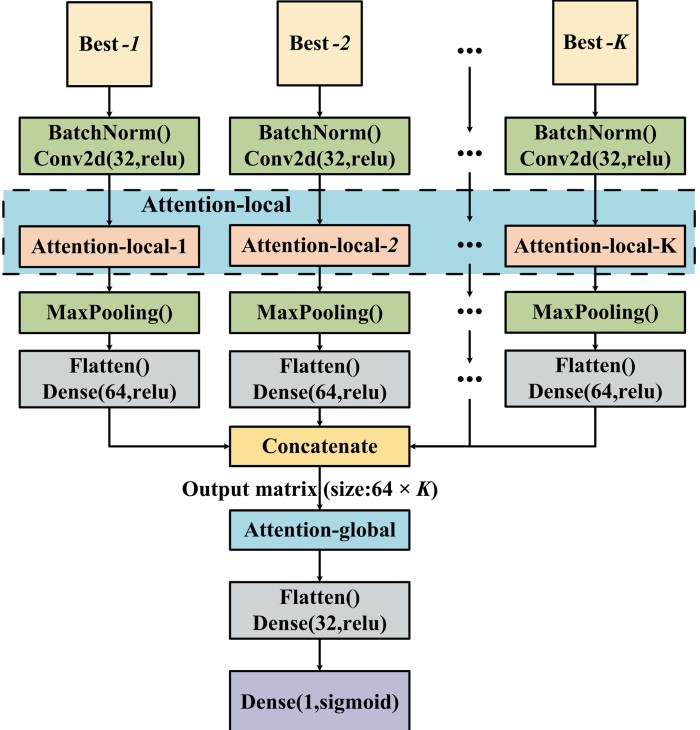

**(a) Architecture of MBC-Attention**

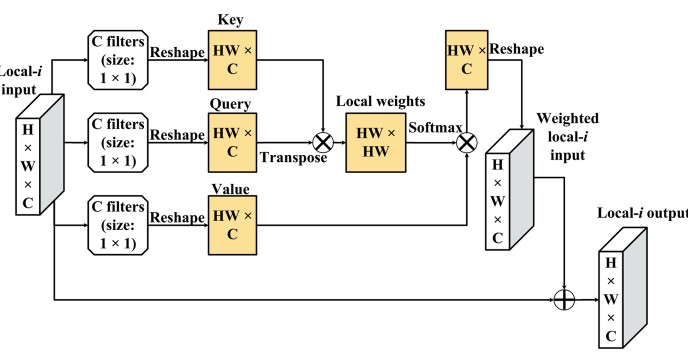

**(b) Attention-local-*i* (*i* ∈ {1, 2, ..., K })**

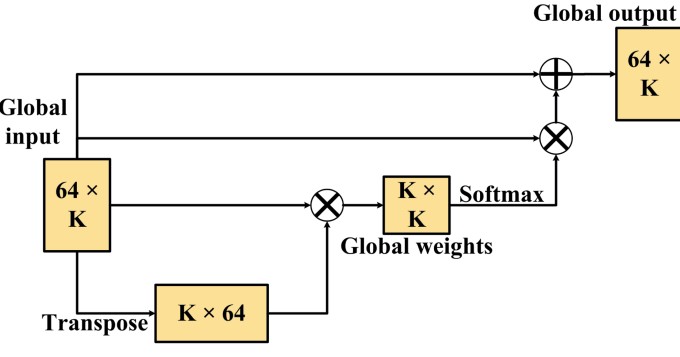

**(c) Attention-global**

**FIG 7** The proposed MBC-Attention method based on (a) the Multi-Branch CNN architecture and enhanced with (b) a local attention and (c) the global attention mechanism.

**TABLE 6** Seventeen traditional machine learning algorithms used in the TML17 method

| Algorithm | Description |
| --- | --- |
| RF | Random Forest regressor (12) |
| ET | Extra Trees regressor (29) |
| LightGBM | Light Gradient Boosting Machine (30) |
| GBR | Gradient Boosting Regressor (9) |
| KNN | K Neighbors regressor (31) |
| DT | Decision Tree regressor (32) |
| BR | Bayesian Ridge (33) |
| LR | Linear Regressor (34) |
| Ridge | Ridge Regressor (10) |
| Huber | Huber regressor (35) |
| Ada | AdaBoost regressor (36) |
| OMP | Orthogonal Matching Pursuit (37) |
| PAR | Passive Aggressive Regressor (38) |
| LAR | Least Angle Regressor (39) |
| Lasso | Lasso Regressor (11) |
| EN | Elastic Net (8) |
| LLAR | Lasso Least Angle Regressor (40) |

## Feature selection

MBC-Attention, which is based on the MBC architecture, is capable to accept multiple feature types as input and independently learn each of them through the use of multiple branch CNN. In order to identify a set of informative feature types for the regressive task, a two-step feature selection was conducted, similar to our prior study (18). In the first step, RF with 400 trees was employed to perform a rapid evaluation of the regressive performance for each feature type. The top-performing feature types were then identified based on the average PCC score of the model. In the second step, MBC-Attention was leveraged to evaluate all of the best-$K$ feature combinations. Here, best-$K$ is the first $K$ feature types from the sorted list returned in the first step. Due to computational constraints, the largest $K$ feature combination that was examined in this work was 26.

## Hyperparameter tuning of MBC-Attention

To find the optimal CNN architecture for MBC-Attention, we conducted a systematic evaluation of model performance. This involved tuning hyperparameters such as the number of convolutional layers (1, 2, and 3), dropout rate (0, 0.1, 0.2, 0.3, 0.4, and 0.5), the number of filters in each layer (16, 32, 48, 64, 80, 96, 112, and 128), and the loss function (MSE and MSLE). The training was done using the epoch of 200, which was found to be adequate. Furthermore, each experiment was repeated three times to ensure the reliability of our results. The parameter search was conducted through grid search and the results were sorted by the average PCC scores on validation. In total, 288 experiments ($=2 \times 3 \times 6 \times 8$) were conducted to find the optimal hyperparameters of the final MBC-Attention model. The final hyperparameters are as follows: CNN layer = 1, filter = 32, dropout = 0.4, and MSLE loss function. To obtain a stable performance in the prediction model, we manually tested the use of an early stopping criterion with different settings for the hyperparameters "patience" and "monitor." We also trained the model using different learning rates, decay rates, and decay steps. We ultimately determined that early stopping monitored with "loss," with a learning rate of 0.0005, patience of 15, decay rate of 0.92, and decay steps of 25 yielded the most optimal combination.

## ACKNOWLEDGMENTS

The authors acknowledge the support of the Government of Canada's New Frontiers in Research Fund (NFRF) (NFRFE-2021-00913). J.L.Y. was supported by the University of Macau (Grant no. MYRG2019-00098-FST) and the Science and Technology Department Fund of Macau SAR (File no. 0010/2021/AFJ). The funders had no role in study design, data collection and interpretation, or the decision to submit the work for publication.

The authors also thank Dr. Pratiti Bhadra for her help in reviewing the paper.

S.W.I.S. and F.X.C.V. conceived the study. J.L.Y. designed the methods, conducted the experiments, analyzed the results, and drafted the manuscript. S.W.I.S., M.Z., and B.Z. supervised the work. S.W.I.S., F.X.C.V., M.Z., and B.Z. finalized the manuscript. All the authors read and approved the final manuscript. S.W.I.S., F.X.C.V., and B.Z. acquired project funding.

The authors declare no conflict of interest.

## AUTHOR AFFILIATIONS

[1]PAMI Research Group, Department of Computer and Information Science, University of Macau, Taipa, Macau, China

[2]School of Computer Science, Chongqing University, Shapingba, Chongqing, China

[3]Host-Microbe Interactions Laboratory, Center for Chemical and Synthetic Biology, Department of Chemistry and Biomolecular Sciences, University of Ottawa, Ottawa, Ontario, Canada

[4]Centre for Infection, Immunity, and Inflammation, University of Ottawa, Ottawa, Ontario, Canada

[5]Department of Biochemistry, Microbiology and Immunology, University of Ottawa, Ottawa, Ontario, Canada

[6]Institute of Science and Environment, University of Saint Joseph, Macau, China

## AUTHOR ORCIDs

Jielu Yan  http://orcid.org/0000-0001-8342-7453
François-Xavier Campbell-Valois  http://orcid.org/0000-0001-5105-2968
Shirley W. I. Siu  http://orcid.org/0000-0002-3695-7758

## FUNDING

| Funder | Grant(s) | Author(s) |
| --- | --- | --- |
| Government of Canada's New Frontiers in Research Fund | NFRFE-2021-00913 | François-Xavier Campbell-Valois |
| University of Macau | MYRG2019-00098-FST | Shirley W I Siu |
| The Science and Technology Department Fund of Macau SAR | 0010/2021/AFJ | Bob Zhang |

## AUTHOR CONTRIBUTIONS

Jielu Yan, Data curation, Investigation, Methodology, Resources, Software, Writing – original draft, Writing – review and editing | Bob Zhang, Project administration, Supervision, Writing – review and editing | Mingliang Zhou, Methodology, Project administration, Supervision | François-Xavier Campbell-Valois, Conceptualization, Funding acquisition, Project administration, Supervision, Writing – review and editing | Shirley W. I. Siu, Conceptualization, Data curation, Formal analysis, Funding acquisition, Investigation, Methodology, Project administration, Resources, Software, Supervision, Validation, Visualization, Writing – review and editing

## DATA AVAILABILITY

All peptide data used in this study were downloaded from DBAASP v3 (30). The data and script files for reproducing the experiments can be downloaded from MBC-Attention. The final predictive models are also available at the above link. The implementation was done using Python 3.9.12. Required libraries and installation instructions are provided in the GitHub page.

## ADDITIONAL FILES

The following material is available online.

### Open Peer Review

**PEER REVIEW HISTORY (review-history.pdf).** An accounting of the reviewer comments and feedback.

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
