## [Reviewer comments · mSystems]

A Deep Learning Method for Predicting the Minimum Inhibitory Concentration of Antimicrobial Peptides against *Escherichia coli* using Multi-Branch-CNN and Attention

Jielu Yan, Bob Zhang, Mingliang Zhou, François-Xavier Campbell-Valois, and Shirley Siu

Corresponding Author(s): Shirley Siu, University of Saint Joseph

Review Timeline:

Submission Date:	April 19, 2023
Editorial Decision:	May 9, 2023
Revision Received:	May 19, 2023
Accepted:	May 31, 2023

Editor: Sergio Baranzini

Reviewer(s): Disclosure of reviewer identity is with reference to reviewer comments included in decision letter(s). The following individuals involved in review of your submission have agreed to reveal their identity: Yuanzhao Ding (Reviewer #1); Didier Barradas-Bautista (Reviewer #2)

Transaction Report:

DOI: <https://doi.org/10.1128/msystems.00345-23>

May 9, 2023

Prof. Shirley W. I. Siu
University of Saint Joseph
Institute of Science and Environment
Macau
China

Re: mSystems00345-23 (A Deep Learning Method for Predicting the Minimum Inhibitory Concentration of Antimicrobial Peptides against *Escherichia coli* using Multi-Branch-CNN and Attention)

Dear Prof. Shirley W. I. Siu:

Thank you for submitting your manuscript to mSystems. We have completed our review and I am pleased to inform you that, in principle, we expect to accept it for publication in mSystems. However, acceptance will not be final until you have adequately addressed the reviewer comments.

Preparing Revision Guidelines

Please return the manuscript within 60 days; if you cannot complete the modification within this time period, please contact me. If you do not wish to modify the manuscript and prefer to submit it to another journal, please notify me of your decision immediately so that the manuscript may be formally withdrawn from consideration by mSystems.

Sincerely,

Sergio Baranzini

Editor, mSystems

Journals Department
Reviewer comments:

Reviewer #1 (Comments for the Author):

Jielu Yan et al. conducted a study on the use of deep learning techniques to predict the minimum inhibitory concentration (MIC) of antimicrobial peptides against *Escherichia coli*. Overall, the paper presents interesting findings and has significant scientific merit. However, there are several areas that require modification before publication.

Major suggestions:

- (1) The authors should provide a brief explanation of the hyperparameter selection process for the machine learning method used, including reference to relevant literature. For example, the authors selected an epoch of 200 and a learning rate of 0.0005 (compared to a common learning rate of 0.001). It would be helpful to know the reasons behind these selections and the primary reference(s) used.
- (2) Considering that the study was conducted with a relatively small dataset of 7,861 data entries, it is necessary to verify whether this sample size is adequate for training the model. The authors should provide additional references to support the reliability of their analysis.
- (3) Given the many factors that influence MIC values for *E. coli* in biological experiments, such as growth media, temperature, and humidity, the authors should exercise caution in their introduction and discussion of the study's findings.

Minor suggestions:

- (1) The authors should provide a brief introduction to the average Pearson correlation coefficient (PCC) in line 107 to aid readers who may not be familiar with this statistical measure. PCC reflects the degree of correlation between two datasets, with a value of -1 indicating a totally negative correlation, 0 indicating no correlation, and 1 indicating a completely positive correlation.
- (2) There may be a typo in Table 5, where "PAAC pseudo-Amino Acid Composition" and "SOCNumber Sequence-Order-Coupling Number" are listed.
- (3) The font size in Figures 1 and 3 (lines 138 and 191) may be too small, and the resolution could be improved to facilitate reading.
- (4) The capitalization of words in Figure 6 (line 419) should be consistent across all images.

Overall, this is a well-written paper, and with the recommended modifications, it would be suitable for publication.

Reviewer #2 (Comments for the Author):

The paper "A Deep Learning Method for Predicting the Minimum Inhibitory Concentration of Antimicrobial Peptides against *Escherichia coli* using Multi-Branch-CNN and Attention" is a natural extension of the authors' previous work. In this paper, they develop a machine learning model that uses deep learning to predict the concentration of antimicrobial peptides meaning that they developed a regression model instead of a binary classifier.

The analysis presented in the paper is robust. The authors show the feature selection process, and they compare the performance metrics of several regression models, including the one they propose. The final comparison tests several modalities of the proposed architecture, showing its versatility and how the user can fine-tune the model to a specialized scope.

The authors show that their proposed model is superior to traditional machine learning algorithms by a small margin. However, it performs better overall.

This is a solid research paper that includes the corresponding checks for a machine learning project.

ASM [mSystems]

Manuscript ID: mSystems00345-23

Type of manuscript: Research Article

Title: A Deep Learning Method for Predicting the Minimum Inhibitory Concentration of Antimicrobial Peptides against *Escherichia coli* using Multi-Branch-CNN and Attention

We sincerely thank the Associate Editor and the anonymous reviewers for their detailed, insightful and valuable comments. We have carefully considered the reviewers' comments and revised the paper accordingly. In the following, please find our responses to each reviewer's comments and a summary of the revisions. For ease of reading, we included the original review comments in a box, followed by our responses.

Please note that:

"main-marked.pdf": the marked-up version of the revised manuscript, in which the deleted text has been shown with a strikethrough and the added text with red color.

Responses to Reviewer #1:

Jielu Yan et al. conducted a study on the use of deep learning techniques to predict the minimum inhibitory concentration (MIC) of antimicrobial peptides against Escherichia coli. Overall, the paper presents interesting findings and has significant scientific merit. However, there are several areas that require modification before publication.

Major suggestions:

AQ:1 = The authors should provide a brief explanation of the hyperparameter selection process for the machine learning method used, including reference to relevant literature. For example, the authors selected an epoch of 200 and a learning rate of 0.0005 (compared to a common learning rate of 0.001). It would be helpful to know the reasons behind these selections and the primary reference(s) used.

Thanks for the suggestion. We have presented the hyperparameter selection procedure in the section "Hyperparameter tuning of MBC-Attention" as shown below.

463 This involved tuning hyperparameters such as the number of convolutional layers (1,
464 2, and 3), dropout rate (0, 0.1, 0.2, 0.3, 0.4, and 0.5), the number of filters in each
465 layer (16, 32, 48, 64, 80, 96, 112, and 128), and the loss function (MSE and MSLE). The
466 training was done using the epoch of 200, which was found to be adequate. Further-
467 more, each experiment was repeated three times to ensure the reliability of our re-
468 sults. The parameter search was conducted through grid search and the results were
469 sorted by the average PCC scores on validation. In total, 288 experiments ($=2 \times 3 \times 6$
470 $\times 8$) were conducted to find the optimal hyperparameters of the final MBC-Attention
471 model. The final hyperparameters are as follows: CNN layer = 1, filter = 32, dropout =
472 0.4, and MSLE loss function. To obtain a stable performance in the prediction model,
473 we manually tested the use of an early stopping criterion with different settings for the
474 hyperparameters "patience" and "monitor". We also trained the model using different
475 learning rates, decay rates, and decay steps. We ultimately determined that early stop-
476 ping monitored with "loss", with a learning rate of 0.0005, patience of 15, decay rate
477 of 0.92, and decay steps of 25 yielded the most optimal combination.

Different data sets use different parameters, and the final choice of hyperparameters is often based on the results of parameter optimization. There are many works dealing with different strategies for selecting the optimal parameters [1, 2]. Overall, the final learning rate is determined based on the validation performance in the training datasets, model architecture, and combination with other hyperparameters. To our understanding and observation, it is not always 0.001.

Moreover, we first stopped at 200 epochs to empirically select the CNN layer, the number of filters, the dropout rate, and the loss function. After we decided on the CNN layer, number of filters, dropout rate, and loss function, we used an early stop function to terminate the training phase and maintain a stable performance. We manually set the parameters of the early stop function and finally determined that the early stop would be monitored with "loss", with a learning rate of 0.0005, a patience of 15, a decay rate of 0.92, and decay steps of 25 as our final selection.

We are aware that there are many different algorithms for hyperparameter optimization [3]. However, since the MBC-Attention model is relatively small, in our case, the grid search is sufficient for this purpose.

[1] Wu, Y., & Liu, L. (2023). Selecting and Composing Learning Rate Policies for Deep Neural Networks. *ACM Transactions on Intelligent Systems and Technology*, 14(2), 1-25.

[2] Wu, Y., Liu, L., Bae, J., Chow, K. H., Iyengar, A., Pu, C., ... & Zhang, Q. (2019, December). Demystifying learning rate policies for high accuracy training of deep neural networks. In *2019 IEEE International conference on big data (Big Data)* (pp. 1971-1980). IEEE.

[3] Yu, T., & Zhu, H. (2020). Hyper-parameter optimization: A review of algorithms and applications. *arXiv preprint arXiv:2003.05689*.

AQ:2 = Considering that the study was conducted with a relatively small dataset of 7,861 data entries, it is necessary to verify whether this sample size is adequate for training the model. The authors should provide additional references to support the reliability of their analysis.

Thanks for the critical suggestion. Just a note on the numbers, the total number of sequences downloaded is 7,861, where the number of sequences finally used after preprocessing is 3,929, with 3,536 sequences forming the training set and 393 sequences forming the test set.

To examine if our sample size is adequate to train the proposed MBC-Attention model, we conducted four experiments trained with randomly selected 1000, 2000, and 3000 sequences from the training set and the full training set (3536 sequences). Each draw-and-run was repeated 5 times, all trained models were tested with the same test set and we measured their performance by PCC. As shown in Figure 4 below, the resulting boxplot of the PCC values exhibit a clear trend of improvement, indicating that the model is able to learn from more training data and make predictions more accurately. The result also shows a clear trend toward convergence, suggesting that the current training data size is a reasonable amount to train the MBC-Attention model. Although the trend is clear, the optimal data size for this AMP prediction task cannot be determined due to unavailability of an even larger data set.

The figure and the analysis are added to the manuscript, page 7-8, as shown below:

252 **Data size effect exploration** Deep learning is data-dependent, i.e., it relies on
253 a sufficient amount of data to build a reliable prediction model. When the amount
254 of data is small, deep learning algorithms often perform poorly (29). Therefore, it is
255 important to decide how much data is needed to train a model to achieve optimal
256 performance and to understand whether the current amount of data is adequate for
257 the training needs. Here, we examined the effects of training data size on model per-
258 formance for the proposed MBC-Attention model. Four experiments were conducted
259 using training sets of 1000, 2000, 3000, and 3536 sequences (the full training set), re-
260 spectively, to train the model. We then evaluated the performance for the model on

FIG 4 Effect of varying training set sizes on the performance of MBC-Attention.

261 same test data set. To ensure statistical significance, each experiment was repeated
262 five times and the results were analyzed.

263 The PCC scores of all experiments on the test set were visualized in a scattered
264 boxplot, as shown in Figure 4. The PCC values show a significant improvement in
265 model performance with an increasing training set size. Specifically, the median PCC
266 values exhibit a clear trend of improvement, indicating that the model is able to learn
267 from more training data and make predictions more accurately. The result also shows
268 a clear trend toward convergence, indicating that the current training data size is a
269 reasonable amount to train the MBC-Attention model. Unfortunately, an even larger
270 data set is currently not available to conclude with the optimal data size for this AMP
271 prediction task.

This analysis result was restated in the discussion part:

324 *E. coli*. Moreover, we examined the effects of different data sizes, and the result shows
325 a good tendency to converge, indicating that the size of our data set is reasonable for
326 training the proposed MBC-Attention model.

AQ:3 = Given the many factors that influence MIC values for E. coli in biological experiments, such as growth media, temperature, and humidity, the authors should exercise caution in their introduction and discussion of the study's findings.

Thank you for this suggestion. We are aware of the different MIC values that can be obtained for the same AMP under different experimental conditions. However, since the dataset is not comprehensive enough to train specific (or condition-dependent) DL models, our approach in this work is to predict the overall antimicrobial potential of an AMP by correlating the sequence with the average MIC from multiple experimental entries.

We improved the manuscript with this idea in the Introduction (page 3) and Method (page 9). We believe that learning the correlation between different experimental conditions and the experimental MIC measure will provide deeper insight on the AMP activity, that should be further investigated in future works (see page 9).

92 predict the MIC value of AMPs against *E. coli* with greater precision. Here, a predicted
93 MIC value refers to the antimicrobial potential of a peptide that is not specific to partic-
94 ular experimental conditions. One should be aware that different MIC values can be
95 obtained for the same AMP under different experimental conditions such as growth
96 media, temperature, pH, ionic strength, and salt type, etc. Although it is noteworthy
97 that standard media is often used to measure MIC in a given bacterial specie, there
98 are other experimental parameters such as the purity of the peptide that can explain
99 diverging reported MIC values for a given AMP. The reason for learning the overall
100 potential instead of a condition-dependent MIC value in the experiment is mainly be-
101 cause there is no comprehensive dataset for AMPs measured under many different
102 conditions. When multiple entries for an AMP exist, the average MIC is calculated as
103 the prediction target. This approach provides a reasonable estimate of the antimicro-
104 bial potential of the sequence. The same approach was adopted in a previous study
105 that developed the PepVAE regression model (7). In this work, MBC-Attention is built

Page 9: Discussion

331 In the future, we plan to extend the application of the MBC-Attention model to
332 predict the antibacterial activity of peptides against other pathogenic bacterial species.
333 To further improve the accuracy of the prediction of experimental MIC for a sequence,
334 it is crucial to investigate the correlation between antimicrobial activity and different
335 experimental conditions. Future research will also focus on investigating this relation-
336 ship and improving our predictive capabilities.

Page 9: Method

344 (μM) and then converted to pMIC by Eq.1. For a sequence that has multiple MIC val-
345 ues, these values were averaged to obtain the mean activity measure for the peptide
346 that represents the overall antimicrobial potential under different experimental condi-
347 tions. Finally, the processed dataset of AMP sequence and MIC for *E. coli* contains 3929

Minor suggestions:

AQ:1 = The authors should provide a brief introduction to the average Pearson correlation coefficient (PCC) in line 107 to aid readers who may not be familiar with this statistical measure. PCC reflects the degree of correlation between two datasets, with a value of -1 indicating a totally negative correlation, 0 indicating no correlation, and 1 indicating a completely positive correlation.

Thank you for the suggestion. We have added the explanation of PCC to the text as shown below (see Page 3).

118 dation, and test sets by repeating the data splitting procedure. For each prediction
119 method, training was performed three times and the method was evaluated based
120 on the average Pearson correlation coefficient (PCC) of the three validation results to
121 determine the best model with optimal parameters. PCC is a measure of linear corre-
122 lation between two datasets that reflects their degree of correlation. The metric with
123 a value of -1 indicates a completely negative correlation, 0 indicates no correlation,
124 and 1 indicates a completely positive correlation. In the comparative study, the pro-

AQ:2 = There may be a typo in Table 5, where "PAAC pseudo-Amino Acid Composition" and "SOCNumber Sequence-Order-Coupling Number" are listed.

Thank you for this comment. We have corrected the wrong spelling as shown below.

SOCNumber	Sequence-Order-Coupling Number (35)	$\lambda \in \{0, 1, \dots, 4\}$
QSOOrder	Quasi-sequence-order (24)	$\lambda \in \{0, 1, \dots, 4\}$
PAAC	Pseudo-Amino Acid Composition (36)	$\lambda \in \{0, 1, \dots, 4\}$

AQ:3 = The font size in Figures 1 and 3 (lines 138 and 191) may be too small, and the resolution could be improved to facilitate reading.

Thank you for the suggestion. The font size in Figure 1 and 3 are enlarged, please refer page 4 and page 6.

AQ:4 = The capitalization of words in Figure 6 (line 419) should be consistent across all images.

Thank you for the suggestion. We have changed the capitalization of the words in Figure 6 and made them consistent across all images as shown in page 13.

Overall, this is a well-written paper, and with the recommended modifications, it would be suitable for publication.

Thank you very much for the affirmation.

Responses to Reviewer #2:

The paper "A Deep Learning Method for Predicting the Minimum Inhibitory Concentration of Antimicrobial Peptides against Escherichia coli using Multi-Branch-CNN and Attention" is a natural extension of the authors' previous work. In this paper, they develop a machine learning model that uses deep learning to predict the concentration of antimicrobial peptides meaning that they developed a regression model instead of a binary classifier.

The analysis presented in the paper is robust. The authors show the feature selection process, and they compare the performance metrics of several regression models, including the one they propose. The final comparison tests several modalities of the proposed architecture, showing its versatility and how the user can fine-tune the model to a specialized scope.

The authors show that their proposed model is superior to traditional machine learning algorithms by a small margin. However, it performs better overall.

This is a solid research paper that includes the corresponding checks for a machine learning project.

Thank you very much for the affirmation.

May 31, 2023

Prof. Shirley W. I. Siu
University of Saint Joseph
Institute of Science and Environment
Macau
China

Re: mSystems00345-23R1 (A Deep Learning Method for Predicting the Minimum Inhibitory Concentration of Antimicrobial Peptides against *Escherichia coli* using Multi-Branch-CNN and Attention)

Dear Prof. Shirley W. I. Siu:

Thank you for the quick and comprehensive revision of your manuscript.

Your manuscript has been accepted, and I am forwarding it to the ASM Journals Department for publication. For your reference, ASM Journals' address is given below. Before it can be scheduled for publication, your manuscript will be checked by the mSystems production staff to make sure that all elements meet the technical requirements for publication. They will contact you if anything needs to be revised before copyediting and production can begin. Otherwise, you will be notified when your proofs are ready to be viewed.

If you would like to submit a potential Featured Image, please email a file and a short legend to msystems@asmusa.org. Please note that we can only consider images that (i) the authors created or own and (ii) have not been previously published. By submitting, you agree that the image can be used under the same terms as the published article. File requirements: square dimensions (4" x 4"), 300 dpi resolution, RGB colorspace, TIF file format.

We recognize that the video files can become quite large, and so to avoid quality loss ASM suggests sending the video file via <https://www.wetransfer.com/>. When you have a final version of the video and the still ready to share, please send it to mSystems staff at msystems@asmusa.org.

Sincerely,

Sergio Baranzini
Editor, mSystems

Journals Department
E-mail: mSystems@asmusa.org